# Impact of physicians' participation in non-interventional post-marketing studies on their prescription habits: A retrospective 2-armed cohort study in Germany

**Cora Koch** [1,2]*, **Jörn Schleeff**[3], **Franka Techen**[3], **Daniel Wollschläger**[4], **Gisela Schott**[5], **Ralf Kölbel**[6], **Klaus Lieb**[2]

**1** Clinic of Neurology and Neurophysiology, Medical Center–University of Freiburg, Faculty of Medicine, University of Freiburg, Freiburg, Germany, **2** Department of Psychiatry and Psychotherapy, University Medical Center Mainz, Mainz, Germany, **3** National Association of Statutory Health Insurance Funds, Berlin, Germany, **4** Institute for Medical Biostatistics, Epidemiology and Informatics, Mainz, Germany, **5** Drug Commission of the German Medical Association, Berlin, Germany, **6** Law Faculty, Ludwig Maximilian University of Munich, Munich, Germany

* cora.koch@uniklinik-freiburg.de

**Data Availability Statement:** Data cannot be shared publicly due to data privacy laws in Germany and the requirement by the German

## Abstract

### Background

Non-interventional post-marketing studies (NIPMSs) sponsored by pharmaceutical companies are controversial because, while they are theoretically useful instruments for pharmacovigilance, some authors have hypothesized that they are merely marketing instruments used to influence physicians' prescription behavior. So far, it has not been shown, to our knowledge, whether NIPMSs actually do have an influence on prescription behavior. The objective of this study was therefore to investigate whether physicians' participation in NIPMSs initiated by pharmaceutical companies has an impact on their prescription behavior. In addition, we wanted to analyze whether specific characteristics of NIPMSs have a differing impact on prescription behavior.

### Methods and findings

In a retrospective 2-armed cohort study, the prescription behavior of 6,996 German physicians, of which 2,354 had participated in at least 1 of 24 NIPMSs and 4,642 were controls, was analyzed. Data were acquired between 6 October 2016 and 8 June 2018. Controls were matched by overall prescription volume and number of prescriptions of the drug studied in the NIPMS in the year prior to the NIPMS. Primary outcome was the relative rate of prescriptions of the drug studied in the NIPMS by participating physicians compared to controls during the NIPMS and the following year. Secondary outcomes were the proportion of prescriptions of the studied drug compared to alternative drugs used for the same indication, the revenue generated by these prescriptions, and the association between the marketing characteristics of the NIPMS and prescription habits. Of the 24 NIPMSs, the 2 largest drug groups studied were antineoplastic and immunomodulatory agents (7/24, 29.2%) and

Ministry of Health to erase the data because of concerns of data privacy. The raw data is held by the GKV-Spitzenverband, the National Association of Statutory Health Insurance Funds, https://www.gkv-spitzenverband.de/, Reinhardtstrasse 28, 10117 Berlin, Germany. Access to the data is subject to data privacy restrictions.

**Funding:** The authors received no specific funding for this work.

**Competing interests:** I have read the journal's policy and the authors of this manuscript have the following competing interests: CK and KL are members of MEZIS, the German "No free lunch"-association. RK, FT, JS and GS state they have nothing to disclose. DW reports grants from German Federal Ministry of Education and Research during the conduct of the study.

**Abbreviations:** DDD, defined daily dose; NIPMS, non-interventional post-marketing study.

agents for the nervous system (4/24, 16.7%). Physicians participating in an NIPMS prescribed more of the studied drug during and in the year after the NIPMS, at a relative rate of 1.08 (95% CI 1.07–1.10; $p < 0.001$) and 1.07 (95% CI 1.05–1.09); $p < 0.001$), respectively. Participating physicians were more likely than controls to prescribe one of the studied drugs rather than alternative drugs used for the same indication (odds ratio 1.04; 95% CI 1.03–1.05). None of the marketing characteristics studied were significantly associated with prescription practices. The main limitation was the difficulty in controlling for confounders due to privacy laws, with a resulting lack of information regarding the included physicians, which was mainly addressed by the matching process.

## Conclusions

Physicians participating in NIPMSs prescribe more of the investigated drug than matching controls. This result calls the alleged non-interventional character of NIPMSs into question and should lead to stricter regulation of NIPMSs.

---

## Author summary

### Why was this study done?

- After drugs are authorized, non-interventional post-marketing studies (NIPMSs) are initiated to study rare side effects or other aspects of the drug that may have been missed during the authorization trials.

- Previous studies have shown that NIPMSs, while conducted regularly, often lack scientific rigor, rarely lead to relevant results, and are rarely published.

- Some authors have therefore hypothesized that NIPMSs primarily serve marketing purposes for pharmaceutical manufacturers by familiarizing physicians with new drugs.

- So far, it is unclear whether NIPMSs actually do have an impact on physicians' prescribing behavior.

### What did the researchers do and find?

- We conducted a study in Germany comparing the prescription behavior of 2,354 physicians who had participated in at least 1 of 24 NIPMSs and 4,642 comparable physicians who had not participated in such studies.

- We found that physicians participating in an NIPMS prescribed 6%–8% more of the drug studied in the NIPMS than comparable physicians during the NIPMS and the year after.

- We also looked at certain characteristics of the NIPMSs to see whether they predicted the impact on prescription behavior, but found no characteristics that were associated with the impact.

**What do these findings mean?**

- NIPMSs seem to have an impact on physicians' prescribing behavior despite their "non-interventional" nature.

- Up to this point, NIPMSs have been only very loosely regulated because it was assumed that they have a low potential to cause harm. However, with the possibility that physicians prescribe differently due to their participation in an NIPMS, which may or may be detrimental to patients, only NIPMSs that are designed to collect essential data should be permitted.

## Introduction

Non-interventional post-marketing studies (NIPMSs) funded by the pharmaceutical industry have been the subject of controversial debate. In principle, their purpose is to provide data on the real-world safety and/or effectiveness of recently authorized drugs by studying them with a larger and less highly selected patient population than is usual in authorization trials [1,2]. Some NIPMSs are imposed by regulatory agencies, for example to assess a safety risk of a medicinal product or to evaluate the effectiveness of risk management measures [3]. However, some authors have hypothesized that many NIPMSs primarily serve marketing purposes for pharmaceutical manufacturers by familiarizing physicians with a newly authorized drug as well as offering an incentive to prescribe the drug [1,4–7].

A recent study of German NIPMSs by Spelsberg et al. showed that they rarely serve to improve drug safety because their sample sizes are usually too small to allow for the detection of rare adverse events [8]. Other studies have also pointed in this direction by showing that NIPMSs usually lack scientific quality [4,5,9]. In addition, Spelsberg and colleagues raised the concern that strict confidentiality clauses in combination with the high remuneration for participation in NIPMSs could actually serve to discourage physicians from reporting adverse events [8]. NIPMSs are also published extremely rarely, even though they are conducted regularly, which makes it even less likely that they will lead to safer prescription practices, even if relevant data are generated [8,10–12]. Another concern is the effect the participation in NIPMSs could have on physician prescription practices. Physicians' participation in an NIPMS usually consists of enrolling patients who are prescribed a certain drug in an NIPMS and gathering data on the enrolled patients regarding parameters such as adverse events or efficacy. For the enrollment of each patient, physicians are remunerated with a certain fee. By remunerating physicians for the inclusion of a patient who is prescribed a certain drug, NIPMSs offer an incentive for the prescription of this drug, possibly affecting prescription practices.

Although several authors have argued that NIPMSs mainly serve the marketing purposes of the pharmaceutical industry, it has not yet been demonstrated, to our knowledge, that the prescription behavior of physicians indeed changes during or after participation in an NIPMS; such an effect has only been demonstrated for interventional studies [13–15]. However, if this were the case, it would be an important reason to increase regulation of such studies. Currently, NIPMSs do not need to be registered in the US; in Germany, they need to be registered, but they do not need to be authorized by a higher federal authority, as interventional clinical trials need to be [16]. Concerns that such studies may not only be less useful than they are

made out to be, but may in addition have deleterious effects on physicians' reporting of adverse events and their prescription practices should lead to stricter scrutiny of such studies before they are initiated.

The objective of the current study was thus primarily to investigate whether participation in NIPMSs impacts the participating physicians' prescription practices. In addition, we wanted to analyze whether such a possible change results in more expensive prescriptions by leading to a shift in prescriptions toward more expensive drugs when there are less expensive alternative drugs used for the same indication. Lastly, we wanted to investigate whether certain characteristics of NIPMSs are useful to predict the impact on the participating physicians' prescription behavior.

## Methods

### Ethics review

In a previous study [9], we gathered data on NIPMSs at the National Association of Statutory Health Insurance Funds (GKV-Spitzenverband), of which some data were used in the current study. The local ethics committee of the Landesärztekammer Rheinland-Pfalz decided that it was not necessary to conduct an ethics review for this previous study. To gain access to the prescription data of physicians participating in NIPMSs as well as controls, we submitted a request to the German Federal Ministry of Health that the National Association of Statutory Health Insurance Funds as well as a specific German statutory health provider (the Innungskrankenkasse) be allowed to provide us with the respective data. This request was the basis for the planning of the study (see S1 File). After this request was granted, we again consulted with the local ethics committee, which again decided that it was not necessary to conduct an ethics review for the current study.

### Study design

In a retrospective 2-armed cohort study, we compared the prescription practices of physicians who had participated in an NIPMS with those of matched controls who had not participated in an NIPMS. After identifying eligible NIPMSs, participating physicians were identified and matched 1:2 to control physicians, resulting in a "matching group" (see Table 1 for eligibility criteria). Prescription data regarding overall prescription volume (i.e., number of packages of all drugs prescribed by a physician) as well as prescriptions of the drug studied in the NIPMS and alternative drugs were acquired for the year before the NIPMS (t0), during the NIPMS (t1), and the following year (t2) (see S1 Appendix for definition of "alternative drug"). Differences in prescription volume during and after the study regarding the studied drugs as well as alternative drugs were used to assess the impact of participation in an NIPMS on prescription practices. In addition, we collected data on the following NIPMS characteristics, which could potentially be indicators of NIPMSs being conducted for marketing purposes: inappropriate remuneration, the medication having been on the market too long, low scientific quality, low formal quality, negligible effort required of physician, missing report regarding the results of the NIPMS, and presence of a secrecy clause in the contract for participating physicians (see S4 Appendix and [9] for further details). These indicators were summarized in a "marketing score" as described in our previous publication, where a higher score indicates a higher likelihood of the NIPMS having been initiated for marketing purposes (see S4 Appendix and [9]). Associations between these characteristics and prescription volume were used to assess whether NIPMSs found to be more likely to have been initiated for marketing reasons had a higher impact on prescription practices.

**Table 1. Inclusion criteria for NIPMSs.**

| Inclusion criteria for NIPMSs | Reason |
|---|---|
| Began after 31 December 2012 | To allow for analysis 1 year prior to the beginning of the NIPMS (t0), as prescription data at the National Association of Statutory Health Insurance Funds were only available beginning in January 2012 |
| Ended after 31 December 2013 | Change in regulations affecting all studies ending after this date that allowed for tracking prescription practices |
| Ended before 1 July 2015 | To allow for a follow up of 1 year (t2) immediately after the end of the NIPMS, while ending before the commencement of our data collection |
| Observed medicinal product or drug is covered by statutory health insurance | Otherwise, notification to the National Association of Statutory Health Insurance Funds is not required regarding participating physicians |
| Observed medicinal product or drug is subject to a prescription by a physician and dispensed by a pharmacy | Otherwise, no prescription data are available |
| Study is prospective | Retrospective studies are not expected to affect prescription practices |
| Observed medicinal product or drug was approved at least 6 months before the beginning of the NIPMS | Otherwise, prescription data before the study would not represent the physicians' prescription habits accurately |
| Study was conducted among physicians in private practice (rather than physicians employed in hospitals) | Prescription data are only available for physicians in private practice |

NIPMS, non-interventional post-marketing study.

## NIPMSs

We considered NIPMSs to be eligible for our study when certain criteria were met that allowed for an assessment of the prescription practices of the physicians between 1 year before and 1 year after the NIPMS. Criteria that enabled these analyses and reasons are given in Table 1.

All physicians participating in one of the eligible NIPMSs who could be identified using the data at the National Association of Statutory Health Insurance Funds and our validation process (see below) and had prescribed a minimum amount overall (to ensure they were still practicing) as well as a minimum amount of the drug studied in the NIPMS were included in the study. We matched 2 controls to each of these physicians using the number of overall prescriptions (in packages) as well as the number of defined daily doses (DDDs) of the studied drug and alternative drugs in the year before the NIPMS began (t0). Due to data privacy laws in Germany, other factors such as physician age, gender, location, and specialization could not be considered for matching. Physicians participating in several different NIPMSs were matched to different controls for each NIPMS (see S2 Appendix for exact process). Two controls were chosen for each case, as opposed to more or fewer, to balance the increase in statistical power through the number of controls with the matching quality given the limited pool of good matches for each case [17,18].

## Data sources and setting

Data regarding the NIPMSs as well as participating physicians were identified from notifications submitted to the National Association of Statutory Health Insurance Funds regarding NIPMSs. In Germany, companies planning such a study need to notify 3 different authorities before its initiation, among others the National Association of Statutory Health Insurance Funds. The notification needs to include location, beginning and end dates, and the objective

of the study as well as a list of participating physicians (by name and by LANR, a unique ID number permanently assigned to each physician) and a study plan. In addition, for studies observing a drug or medicinal product that is covered by statutory health insurance, companies need to provide information regarding type and amount of remuneration received by the participating physicians as well as a sample contract between the company and the physicians. The amount of remuneration needs to be justified by the effort required by the physician, and this justification must be described by the company in the notification. Data regarding the characteristics of the studies as well as the identifying data on physicians were gathered between 6 October 2016 and 9 January 2018. Because notifications were often incomplete or faulty regarding the identifying information on physicians, we conducted a validation by comparing the acquired data with a directory at the Innungskrankenkasse containing all practicing physicians in Germany. This validation process took place between 22 and 26 January 2018. See S3 Appendix for a precise description of this process.

Prescription data were acquired using data from the GKV-Arzneimittelschnellinformation (GAmSi) project, which consists of data reported to the National Association of Statutory Health Insurance Funds by pharmacies regarding filled prescriptions for patients with statutory health insurance (in 2018, 87% of German citizens were insured by one of the statutory health insurance providers [19]). For each physician, we acquired data regarding the drug studied in the NIPMS they participated in or were matched to as controls, as well as alternative drugs and overall number of prescriptions before, during, and after the NIPMS. Prescription data were enriched with the official version of the German Anatomical Therapeutic Chemical (ATC) classification and DDD published by the Wissenschaftliches Institut der AOK (WIdO, Version 49, 201803) [20]. Matching and acquiring the prescription data took place between 27 January and 8 June 2018.

## Outcome measures

The primary outcome was the relative rate of prescriptions of the drug studied in the NIPMS by participating physicians compared to their respective controls during the NIPMS and the year after.

Secondary outcomes were the proportion of prescriptions of the drug under study compared to alternative drugs, as well as the revenue generated by these prescriptions. In addition, the association between the marketing characteristics of the NIPMS and prescription habits was a secondary outcome.

## Bias

We used directed acyclic graph modeling to identify the minimal sufficient adjustment set of covariates in the regression model to achieve unbiased estimation of the putative causal effect of participation in an NIPMS on prescription volume at t1 [21]. If confounders have a direct effect on the exposure, but only an indirect effect on the outcome via prescription volume at t0, then matching on prescription volume at t0 is sufficient for unbiased estimation (see S1 Fig for the directed acyclic graph model). Other confounders could only introduce bias through a direct effect on prescription volume at t1. Such a confounder would have to act differently on prescription volume at t1 compared to its effect on prescription volume at t0. This would seem to be implausible for candidate confounders such as age, gender, and specialization of physicians that could not be considered due to data privacy laws in Germany.

To account for confounding by physician prescription habits (in general as well as of the drug studied in the NIPMS) before entering into the NIPMS, we used prescription metrics as matching parameters. We assumed prior prescription behavior for the studied drug to be a

good indicator of the previous interest of the physicians in the studied drug. In addition, in the statistical analysis, results were adjusted for prescription practices before entering the NIPMS as well as for total prescription volume during and after the NIPMS.

## Sample size

Because there are no prior studies to our knowledge of changes in prescription practices due to NIPMSs, we had no plausible effect size estimate, and were therefore not able to calculate a target sample size. We thus aimed to include all studies within the time frame with available prescription data that matched our inclusion criteria.

## Statistical methods

Matching groups without a participating physician and without at least 1 control were excluded from data analysis. Mean number of package prescriptions and DDDs were first calculated within each NIPMS, and then averaged across NIPMSs, weighted by the number of contributing matching groups within each NIPMS. Matching-group-wise differences were first averaged within each NIPMS, and then averaged across NIPMSs, weighted by the number of contributing matching groups within each NIPMS. The weighting scheme was used to ensure that more reliable estimates based on NIPMSs with more matching groups had more influence than more uncertain estimates based on NIPMSs with fewer matching groups.

Conditional Poisson regression was used to assess the relative rate of prescriptions for the studied drug in participating physicians versus their respective controls [22]. Coefficients for matching groups were treated as nuisance variables and were eliminated. To account for possible overdispersion, a quasi-Poisson approach was chosen. The log number of days of the NIPMS was used as offset. Regression models for $t0$ were adjusted for the total number of prescriptions during $t0$. Regression models for $t1$ and $t2$ were adjusted for the number of prescriptions for the studied drug during $t0$, and for the total number of prescriptions during $t1$ and $t2$, respectively.

Mixed binomial logistic regression was used to assess a shift in the proportion of prescriptions of studied drugs relative to alternative drugs used for the same indication in participating physicians versus controls. The model included a random intercept effect for matching group and allowed for 0 inflation since a relevant number of matching groups had 0 prescriptions for the drug under study. This analysis was only carried out for $t1$, in the interest of parsimony.

Logistic regression using generalized estimating equations was used to assess the association of marketing indicators with the probability that a participating physician prescribed more of the drug under study than the average of the corresponding controls. Clusters were defined by NIPMS, with the assumption of compound symmetry for the correlation structure.

$p$-Values less than 0.05 were considered statistically significant. Data were analyzed using the R environment for statistical computing version 3.5.2 with packages gnm, brms, and geepack [23–26].

## Sensitivity analyses

In general, we assumed that physicians had participated in an NIPMS for the entirety of the study, even though it is likely that some physicians were recruited for participation after the NIPMS had begun or stopped participating before it was officially terminated. For a subset of physicians, a more precise time frame of participation could be inferred; this was the case when NIPMS sponsors regularly reported on the participating physicians, and it was possible to determine when a specific physician first entered the study and when they stopped participating. Sensitivity analyses were conducted for this subset of physicians. In addition, sensitivity

analyses were conducted that included only the drug manufactured by the sponsor of the study, as generics were available for some of the drugs studied in the NIPMSs.

## Results

### Study population

Of a total of 95 registered NIPMSs that began and ended in the predefined time frame, 24 matched our inclusion criteria, and 2,354 physicians that had participated in those NIPMSs could be analyzed (See S1 Fig and S2 Table for information on exclusion of physicians and NIPMSs, respectively). The mean duration of NIPMSs was 500 days (SD 181 days). The mean marketing score was 2.4 (SD 1.5; range 0–5, maximum possible value 7.5) (see Tables 2 and S2 for characteristics of individual NIPMSs). For 1,286 physicians participating in 9 NIPMSs, we could define a more specific time frame of participation in the NIPMS to conduct sensitivity analyses.

### Matching quality

There was an average relative difference of 0.2% for overall number of prescribed packages and −0.83% for overall DDDs between cases and controls during time period t0. There was an

**Table 2. List of NIPMSs and selected characteristics.**

| NIPMS | Studied substance(s) | Number of participating physicians | Start date | Duration (days) | Marketing score |
|---|---|---|---|---|---|
| 1 | Fluocinolone acetonide | 1 | 5 Nov 2013 | 529 | 1.5 |
| 2 | Mometasone | 2 | 3 Feb 2014 | 210 | 0.5 |
| 3 | Paclitaxel | 4 | 1 Mar 2013 | 731 | 5 |
| 4 | Telaprevir | 5 | 6 May 2013 | 756 | 1.5 |
| 5 | Sorafenib | 7 | 16 Jul 2013 | 708 | 2.5 |
| 6 | Filgrastim/pegfilgrastim | 7 | 24 Jan 2013 | 397 | 3 |
| 7 | Infliximab/golimumab | 16 | 18 Mar 2013 | 823 | 3.5 |
| 8 | Tapentadol | 16 | 1 Apr 2013 | 609 | 3.5 |
| 9 | Darbepoetin alfa | 19 | 15 Jan 2013 | 775 | 5 |
| 10 | Docetaxel | 21 | 1 Mar 2013 | 731 | 5 |
| 11 | Denosumab | 40 | 31 Jan 2013 | 415 | 2.5 |
| 12 | Infliximab | 43 | 23 Jan 2013 | 555 | 2.5 |
| 13 | Ciclosporin | 46 | 1 Jan 2014 | 546 | 2 |
| 14 | Iron (III) isomaltoside | 49 | 1 May 2013 | 396 | 4.5 |
| 15 | Rasagiline | 65 | 27 Jan 2014 | 339 | 0 |
| 16 | Rivastigmine | 66 | 15 Apr 2013 | 657 | 1.5 |
| 17 | Fluorouracil and salicylic acid | 117 | 15 Jan 2014 | 410 | 1 |
| 18 | Propiverine | 149 | 31 Jul 2014 | 335 | 0.5 |
| 19 | Ingenol mebutate | 171 | 15 Jul 2013 | 351 | 0 |
| 20 | Agomelatine | 219 | 1 Mar 2014 | 245 | 2.5 |
| 21 | Testosterone | 220 | 6 Oct 2014 | 268 | 3.5 |
| 22 | Timolol and bimatoprost | 281 | 25 Nov 2013 | 402 | 3.5 |
| 23 | Ivabradine | 312 | 17 Mar 2014 | 458 | 2.5 |
| 24 | Olodaterol/tiotropium bromide | 478 | 1 Jun 2014 | 365 | 0.5 |

NIPMSs for which a more precise time frame of participation could be defined and that were therefore used in sensitivity analyses (see Methods) are marked in gray.
NIPMS, non-interventional post-marketing study.

**Table 3. Mean number of prescriptions of the studied drug for controls and participating physicians during the mentioned time frame, weighted by the number of matching groups in the NIPMS.**

| Time frame | Mean duration (days) | Number of packages | | DDD | |
|---|---|---|---|---|---|
| | | Control | NIPMS | Control | NIPMS |
| t0 | 365 | 102.4 | 103.4 | 7,705 | 7,753 |
| t1 | 500 | 109.5 | 120.2 | 8,064 | 8,700 |
| t2 | 365 | 102.1 | 110.3 | 7,736 | 8,314 |

Note that t1 is not the same duration as t0 and t2; number of prescriptions may be compared between groups during the same time frame, but not between time frames.

Control = matched physicians, $n$ = 4,642; NIPMS = physicians participating in an NIPMS, $n$ = 2,354.

DDD, defined daily dose; NIPMS, non-interventional post-marketing study.

average relative difference of 8.29% for number of packages and 10.13% for number of DDDs prescribed of the studied drug between cases and controls during time period t0.

## Primary outcome

Participating physicians showed consistently higher absolute prescription volumes of the studied drug compared with controls, with the gap widening during t1 and narrowing during t2 (see Table 3). Relative to physicians not participating in an NIPMS, physicians participating in an NIPMS had a 7%–8% higher prescription rate of the studied drug during the NIPMS and a 6%–7% higher prescription rate during the year after the NIPMS had finished, when accounting for their overall prescription volume during the respective time frame as well as the number of prescriptions of the studied drug before the start of the NIPMS (see Table 4).

Sensitivity analyses of the relative prescription rate considering only studied drugs manufactured by the sponsor (i.e., excluding generics) showed similar results, as did analyses using only data from physicians for whom a more precise time frame of participation in the NIPMS could be determined. However, in this smaller group, the difference was not statistically significant for the time period after the NIPMS (t2) regarding DDD (see S3–S5 Tables for exact data).

## Secondary outcomes

**Shift in proportion of studied drugs and financial effect.**   The odds of a participating physician prescribing a drug studied in an NIPMS rather than an alternative drug used for the same indication during the time frame of the NIPMS (t1) was slightly higher than for the

**Table 4. Estimated relative prescription rate (Rpr) of the studied drug for participating physicians versus controls.**

| Time frame | Number of packages | | DDD[*] | |
|---|---|---|---|---|
| | Rpr[**] (95% CI) | p-Value | Rpr (95% CI) | p-Value |
| t0 | 1.04 (1.03–1.05) | <0.001 | 1.04 (1.03–1.04) | <0.001 |
| t1 | 1.08 (1.07–1.10) | <0.001 | 1.07 (1.06–1.09) | <0.001 |
| t2 | 1.07 (1.05–1.09) | <0.001 | 1.06 (1.04–1.08) | <0.001 |

Model for t0 adjusted for overall prescription rate; models for t1 and t2 adjusted for overall prescription rate and prescription rate of studied drug at t0.

[*]Defined daily dose (DDD) of the drug studied in the non-interventional post-marketing study.

[**]Relative rate; $n$ = 2,354 groups.

**Table 5. Number of NIPMSs fulfilling each of the marketing indicators.**

| Marketing indicator | Number (%) of NIPMSs* |
|---|---|
| Remuneration was inappropriate or not clearly warranted | 10 (41.7%)** |
| Drug has been on the market for too long | 8 (33.3%) |
| Low scientific quality | 10 (41.7%) |
| Low formal quality | 5 (20.8%) |
| Negligible effort required by physician | 1 (4.2%) |
| Required report missing | 7 (29.1%) |
| Contract contains a secrecy clause | 13 (54.2%) |

Marketing indicators described in [9] and S4 Appendix.

*NIPMSs fulfilling the characteristics; $n = 24$.

**Not appropriate, 4 (16.7%); unclear whether appropriate, 6 (25.0%).

NIPMS, non-interventional post-marketing study.

controls (odds ratio 1.04; 95% CI 1.03–1.05; $p < 0.001$). The mean revenue generated from prescriptions of NIPMS and alternative drugs during this time frame, i.e., the amount that needed to be reimbursed by statutory health insurance providers for prescriptions written by the physicians during t1, was also higher for participating physicians, with a mean revenue of 226,713€ (SD 989,087; median 32,036€) versus 153,013€ (SD 615,225; median 25,521€) for controls. However, this difference was not statistically significant, with an odds ratio for participating physicians to generate a higher revenue than controls of 1.03 (95% CI 0.98–1.08; $p = 0.18$).

**Marketing indicators.** NIPMSs scored a mean of 2.42 (SD 1.54) on the marketing score. See Table 5 for details on how many NIPMSs fulfilled each of the marketing indicators. None of the marketing indicators were significantly associated with prescription volume of the studied drug by physicians participating in an NIPMS compared to controls. While a negligible effort required for the participating physicians was associated with a lower impact on prescription volume in the univariate analysis, this difference was not confirmed by multivariate analysis. See Tables 6 and 7 for results of univariate and multivariate regression analyses, respectively.

## Discussion

This study is the first to our knowledge to show in a quasi-experimental design that physicians participating in an NIPMS show changed prescription rates in favor of the investigated drug.

**Table 6. Results of univariate regression analysis regarding the relationship between marketing characteristics and probability that participating physicians prescribe more of the studied drug than controls.**

| Marketing characteristic | Odds ratio (95% CI) |
|---|---|
| Remuneration is inappropriate or not clearly warranted | 1.01 (0.79–1.30) |
| Drug has been on the market for too long | 0.86 (0.62–1.20) |
| Low scientific quality | 1.03 (0.80–1.33) |
| Low formal quality | 0.91 (0.66–1.25) |
| Negligible effort required by physician | **0.77 (0.68–0.88)** |
| Required report missing | 0.98 (0.67–1.42) |
| Contract contains a secrecy clause | 1.15 (0.90–1.47) |
| Marketing score | 1.00 (0.91–1.08) |

Marketing indicators described in [9] and S4 Appendix. Significant difference in bold.

**Table 7. Results of multivariate regression analysis regarding the relationship between marketing characteristics and probability that participating physicians prescribe more of the studied drug than controls.**

| Marketing characteristic | Odds ratio (95% CI) |
|---|---|
| Remuneration is inappropriate or not clearly warranted | 1.19 (0.65–2.17) |
| Drug has been on the market for too long | 0.80 (0.52–1.25) |
| Low scientific quality | 1.03 (0.82–1.28) |
| Low formal quality | 1.06 (0.84–1.34) |
| Negligible effort required by physician | 0.71 (0.50–1.01) |
| Required report missing | 0.93 (0.49–1.77) |
| Contract contains a secrecy clause | 1.04 (0.76–1.41) |

Marketing indicators described in [9] and S4 Appendix.

Physicians participating in an NIPMS had a meaningfully higher rate of prescription of the studied drug both during and a year after the NIPMS, with an increase of 6%–8%, though the impact was slightly smaller after the NIPMS had ended. In addition, they were more likely to prescribe the studied drug rather than alternative drugs used for the same indication during the NIPMS. This led to a tendency (albeit non-significant) toward higher revenue being generated for pharmaceutical companies by participating physicians' prescriptions. None of the marketing indicators (i.e., indicators of NIPMSs being used for marketing purposes) proposed by our group in an earlier study were useful to predict whether an NIPMS would have a larger or smaller impact on prescription practices [9].

The reasons for the difference in prescription habits between participant physicians and controls cannot be assessed in this study. The difference may be due to a higher awareness of the studied drug because of participation in the NIPMS. Whether the remuneration offered for the inclusion of patients in the study plays a role is unclear, but the amount of remuneration and whether it is appropriate with respect to the amount of effort for the physician does not seem to be associated with the difference. The prescription of the studied drugs also increased compared to alternative drugs used for the same indication. This suggests that the difference is not due to increased diagnosis of the disease that the drug is used to treat, but rather due to a shift in prescription behavior towards the studied drug. Thus, patients with similar disorders are likely to be treated differently by a physician participating in an NIPMS compared to one not participating in an NIPMS. Although this study did not attempt to assess the appropriateness of medical prescriptions, the data nonetheless raise questions about the independence of physicians when prescribing drugs. Physicians participating in NIPMSs showed a higher prescription rate of the drug under study even before the start of the study, even though this was one of the matching criteria. We believe this may be due to the fact that physicians participating in an NIPMS may have already been in contact with representatives of the sponsor of the NIPMS and thus may have already been more aware of the drug compared to controls.

## Strengths and weaknesses

One strength of this study is its large sample size, studying close to 7,000 physicians within a diverse collection of NIPMSs. It is thus highly likely that the results are generalizable to other NIPMSs and other physicians. The study design was quasi-experimental, allowing for assessment of causality when certain assumptions are met. However, controlling confounders was difficult due to data privacy laws in Germany. It is unclear whether physician age, gender, or specialization may have an influence on prescription habits. However, as mentioned in the

Methods section, we used directed acyclic graph modeling to assess the effects of confounders, and believe it is unlikely we insufficiently controlled for these characteristics. Only in rare cases where a studied drug gains an indication during the time of the NIPMS could specialization lead to additional confounding, when one specialty would prescribe the studied drug for the new indication while another specialty would not. One other confounder that may not be sufficiently controlled for is that physicians may have chosen to participate in an NIPMS because they were already aware of a certain medication and actively wanted to gather experience using it, leading to an overestimation of the impact. In our view, however, this is not very likely because pharmaceutical companies are more likely to recruit physicians for NIPMSs who are not as enthusiastic about their medication yet [27].

Another weakness is the imprecise definition of the time frame of physician participation in the NIPMSs. Due to a lack of information regarding when exactly a physician entered or exited an NIPMS, we assumed in the primary analysis that all physicians had participated for the entire time of the NIPMS. This may lead to an under- or overestimation of the difference where we miscalculated the time frame of participation. However, our sensitivity analyses with the subset of physicians for whom a more precise time frame could be determined confirmed the difference we found in the larger set of physicians. It is thus unlikely that the difference would be changed substantially if precise data were available for all physicians.

## Relation to other studies

To our knowledge, so far no other study has studied the impact of NIPMSs on participants' prescription practices. Previous studies have focused primarily on the scientific quality of NIPMSs or the quality of the registrations [4,5,8]. The comparison with trials investigating interventional studies' effects on prescription practices is difficult. For 2 trials identified as seeding trials, quantitative data are not available in sufficient quality to compare with the results of our current study [13,14]. Andersen et al. conducted an independent study of the effects of an interventional trial on physicians' prescription practices and found increases in prescription habits comparable with those found in our current study, though slightly larger [15]. Due to the interventional nature of the trial, it is not surprising that it may have had a more pronounced impact on prescription practices than the NIPMSs in our study. Glass studied relative grant amounts from pharmaceutical companies to physicians participating in phase III trials and found no correlation between the relative grant amount and the subsequent prescription behavior of participating physicians, in line with our result that the appropriateness of remuneration for the NIPMS was not associated with the difference in prescription behavior [28].

## Meaning and implications

Our study shows that participating in a "non-interventional" study may still lead to a change in prescription behavior of the participating physicians. This adds to the large body of evidence indicating that conflicts of interest resulting from interactions between physicians and the pharmaceutical industry influence physician behavior [29–33]. It is unclear whether the change in prescription behavior resulting from NIPMSs is in the best interest of the patient, but currently neither the physicians nor the patients participating in such studies are being informed about it at all. More importantly, NIPMSs are currently subject to less scrutiny than interventional trials due to the assumption that they do not impact physicians' prescriptions and thus do not result in patients being treated differently; for example, it is not required to acquire informed consent from a patient before enrolling them in an NIPMS. Our study casts strong doubts on this assumption. In addition, we were not able to show that certain

marketing characteristics of an NIPMS are able to predict whether it impacts physician prescribing behavior. We have to thus assume that it will not be possible to regulate NIPMSs in a way that reduces their impact on prescribing behavior. This leads to our conclusion that NIPMSs should only be permitted when they are imposed by regulatory authorities or registered with a scientifically sound study design that allows for the collection of essential data.

## Supporting information

**S1 Appendix. Selection of alternative drugs.**
(DOCX)

**S2 Appendix. Matching methods.**
(DOCX)

**S3 Appendix. Validation process.**
(DOCX)

**S4 Appendix. Marketing.**
(DOCX)

**S1 Fig. Directed acyclic graph model of possible confounders.**
(TIF)

**S2 Fig. Flowchart of exclusion of participating physicians and controls.**
(TIF)

**S1 File. Request to German Federal Ministry of Health.** Request regarding the use of privacy-protected data for the purpose of this study.
(DOCX)

**S2 File. STROBE statement.**
(DOC)

**S1 Table. Reasons for exclusions of NIPMSs.**
(DOCX)

**S2 Table. Detailed information on characteristics of NIPMSs.**
(DOCX)

**S3 Table. Sensitivity analysis 1.** Relative prescription rates of participating doctors versus controls considering only studied drugs manufactured by the sponsor (model for t0 adjusted for overall prescription rate; models for t1 and t2 adjusted for overall prescription rate and prescription rate of studied drug at t0).
(DOCX)

**S4 Table. Sensitivity analysis 2.** Relative prescription rates of participating doctors versus controls using only data for physicians with a precise time frame of NIPMS participation (model for t0 adjusted for overall prescription rate; models for t1 and t2 adjusted for overall prescription rate and prescription rate of studied drug at t0).
(DOCX)

**S5 Table. Sensitivity analysis 3.** Relative prescription rates of participating doctors versus controls using only data for physicians with a precise time frame of NIPMS participation and considering only drugs manufactured by the sponsor (model for t0 adjusted for overall prescription rate; models for t1 and t2 adjusted for overall prescription rate and prescription rate

of studied drug at t0).
(DOCX)

**S6 Table. Alternative drugs for each NIMPS.** Note that only alternative drugs that shared the first 3 places of the ATC code with the studied drug were included. For some studied drugs, alternative medications with a different ATC code may exist. In addition, for some alternative drugs, the dosage form was restricted to ensure comparability.
(DOCX)

## Acknowledgments

We thank Andrea Appel, Ellen Gugelfuß, and Fide Marten for support in the data collection phase of the study. We thank Christian Fischer at the Innungskrankenkasse for help in the organization of the data validation process.

## Author Contributions

**Conceptualization:** Cora Koch, Gisela Schott, Ralf Kölbel, Klaus Lieb.

**Data curation:** Jörn Schleeff, Franka Techen, Daniel Wollschläger.

**Formal analysis:** Cora Koch, Daniel Wollschläger.

**Methodology:** Cora Koch, Jörn Schleeff, Franka Techen, Daniel Wollschläger, Ralf Kölbel, Klaus Lieb.

**Project administration:** Cora Koch, Ralf Kölbel, Klaus Lieb.

**Software:** Jörn Schleeff, Daniel Wollschläger.

**Supervision:** Cora Koch, Ralf Kölbel, Klaus Lieb.

**Writing – original draft:** Cora Koch.

**Writing – review & editing:** Cora Koch, Jörn Schleeff, Franka Techen, Daniel Wollschläger, Gisela Schott, Ralf Kölbel, Klaus Lieb.

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
