## [Decision Letter · Decision Letter 0]

24 Mar 2020

Dear Dr. Koch,

Thank you very much for submitting your manuscript "Effect of physicians’ participation in non-interventional post-marketing studies on their prescription habits: A retrospective two-armed cohort study" (PMEDICINE-D-19-04307) for consideration at PLOS Medicine. 

[LINK]

In light of these reviews, I am afraid that we will not be able to accept the manuscript for publication in the journal in its current form, but we would like to consider a revised version that addresses the reviewers' and editors' comments. Obviously we cannot make any decision about publication until we have seen the revised manuscript and your response, and we plan to seek re-review by one or more of the reviewers. 

We expect to receive your revised manuscript by Apr 14 2020 11:59PM. Please email us (plosmedicine@plos.org) if you have any questions or concerns.

We look forward to receiving your revised manuscript. 

Sincerely,

Clare Stone, PhD

Managing Editor 

PLOS Medicine

plosmedicine.org

Abstract – Please add dates of the study as well as geographic information. I would usually ask for summary demographic information to be added to the abstract and it isn’t really fully relevant here, but I still think some information like average age and sex ratio would be good. Also Please be more explicit about what the study’s limitations are as the final sentence of the ‘Methods and Findings’ section. 

Data – even though data can’t be shared by you, please provide the email or link where you requested the data from so others can do the same. 

Refs in main text - Please use square brackets instead of rounded. 

Formatting errors/ - Line 99, for example. There appear to be errors in the main text )Error! Bookmark not defined.. 

Did your study have a prospective protocol or analysis plan? Please state this (either way) early in the Methods section.

Please complete the CONSORT checklist and ensure that all components of CONSORT are present in the manuscript, including [how randomization was performed, allocation concealment, blinding of intervention, definition of lost to follow-up, power statement].

Comments from the reviewers:

Reviewer #1: The authors report the findings of a quasi-experimental study evaluating the effects of non-interventional post-marketing studies on prescribing volume rate of prescribing of the particular drug studied in the NIPM during and in the following year. They matched every intervention physician with 2 controls who were not participants in an NIPM study. 

The study found that the relative rate of prescribing of physicians in NIPM studies were higher for particular drug studied during and in the year after. 

The strengths of the study include its large study sample, attempt to control for confounding through matching by prescription volume of prescriptions and defined daily dosage in the year before, and the use of DAGs to conceptualise the potential causal pathways. 

The use of a conditional Poisson regression analysis to determine the relative rates is appropriate type of statistical model for this type of analysis with an outcome related to count data. 

Major comments

The primary weaknesses are due to its non-randomised design, only matching based on a very limited number of factors (due to availability of data due to privacy laws), confounding by indication and residual confounding (e.g. propensity for certain characteristics of physicians who may join a NIPM study in the first place are not taken into account), lack of a sample size calculation on a potential hypothesized effect size. 

The authors could have also considered, for the experimental arm (physicians involved in NIPMS), a negative exposure drug which was the studies drug under question and conduct a sensitivity analyses to look at the relative rate of a non-studied drug compared to matched controls. Critically, if there still is an effect, it is likely that there will be residual confounders which were not considered in the primary analyses. If there is no effect - this would strengthen the conclusion of the paper. If there is an effect - unfortunately this would make their findings less than definitive. 

Due to this - my feeling was the conclusion was too definitive given the limitations in the non-randomised design of the study as well as the short-time horizon (for instance, does this effect persist after the NIPM ends: the results in Table 3 show that at t1 the prescribing increases but then substantially falls towards t2 at 1 year). The authors' primary conclusion that NIPMS should be more strictly regulated due to changes in prescribing practices in favour of the investigated drug. However, the key issue that hasn't been investigated is whether any of these changes prescribing practices actually result in clinical benefit, clinical harm, or no change for the patient. 

Minor comments: 

Line 99 - check reference link, seems to be an error

Sample size calculation: There must be some rationale for 1:2 matching or else why not just run the analysis on 1:1 matching. In the discussion, the authors mention a previous trial (ref 15) Anderson et al. which did seem to have a measure of increase % of sales in an interventional trial. Could the authors not have used the trials figures to give an estimated samples size powered on an expected difference from the cited trial?

Table 3 results - Could the authors explain why there an increase for both control and NIPMS groups during t1 and the residual effects seems to reduce over time for both groups

Line 341 - Statement is factual incorrect - The study itself was not a trial design and quasi-experimental trials are not designed to specifically assess causality but rather this type design allows you to use casual inference methods when RCT are infeasible. 

Lines 356 - Authors touch on this issue in the discussion, but how you deal confounding by indication that those physicians who also participate in NIPMs are more likely to sign up to medications may be interested in or have a preference for? One suggestion could be to consider a negative control drug (which is not studied in the NIPMS) for participating physician. 

Reviewer #2: This is a fascinating article written on a highly important subject. The influence of industry in medical practice is of great concern globally. One particular way in which researchers have suspected that industry might try to influence physicians is through the conduct of post-marketing trials, which are ostensibly about generating new scientific information but are hypothesized to serve primarily marketing or "seeding" purposes. However, due to lack of uniform data across different companies, trials, and health care institutions, this hypothesis has been difficult to test. The authors of this study are to be commended for their application of a novel data source which appears to be exactly the kind of data needed to test this hypothesis. 

I have several suggestions below to improve the manuscript. I will divide these into "major" and "minor" comments; in my view "major" comments are those that I would consider mandatory to be adequately addressed before publication. "Minor" comments are recommended, but may not be essential. 

Major recommendations:

1. The manuscript is not clearly written. It would greatly benefit from the assistance of a medical writer to help edit for flow, style, and clarity. There are also several places in the manuscript, particularly the Methods, which need to be reorganized in order to convey more clearly what analyses were done. 

2. Related to comment (1) above, I am still not 100% sure that I have fully understood the analytic approach that was taken in this paper. The analysis appears to be strong, but I would want to review the methodology again after it has been edited for clarity in order to make sure that this is the case. 

3. Line 127 refers to "eligible NIMPS," which as of this point has not been defined yet. I would suggest starting the methods section with a full and clear description of inclusion of NIMPS before launching in to other details. This is in line with comment #1 above regarding need to reorganize the Methods section for greater clarity. 

4. The analysis of prescribing during t1 needs to be described more fully. Periods t0 and t2 are both presented as being exactly 1 year in length. But t1 is not, and will be different for every NIMPS. Were prescriptions during t1 annualized? If so, did the authors account for uncertainty in estimates arising from shorter NIMPS being "inflated" up to a full year period? 

5. Related to #4 above, the authors refer to prescribing "rates," but the results I see appear to be simply prescribing counts. A rate implies a "per unit time" in the measure. Are the results presented in Table 3 (for example) the number of prescriptions PER YEAR? If so, this would in fact be a rate…but this should be described more clearly. 

6. More detail needs to be given regarding the "indicators of marketing." Because of the centrality of these measures to the current study, I don't think it is enough to refer only to the prior publication. It would be fine to put it in the supplement, but I think more description (are these measures objective? Subjective? What criteria were they based on?) needs to be present in the current manuscript. 

7. The authors need to include results regarding the success of their matching algorithm. The central question that needs to be addressed here is how closely the exposed and control physicians were able to be matched on their prescription counts. For example, the difference in the number of prescriptions between the exposed and control physicians could be determined within each matched trio, and then the distribution of the difference across all matched trios could be shown in a supplementary table. Without presentation of data such as these (or some other analysis to the same effect), the apparent difference between exposed and control physicians in terms of drug prescribed during the t0 period is highly concerning. 

8. I believe that physician specialty has a potential role as a confounder beyond that which is discussed by the authors. For some drugs, physician specialty could affect prescriptions at t1 independent of the effect at t0; this is because drugs are constantly gaining new indications and usages which may apply to only some specialties. For example, infliximab has longstanding uses for rheumatologic/inflammatory conditions, but increasing use in treating immunotherapy-related toxicity in cancer patients. Therefore, I would expect t0 = t1 for a rheumatologist or gastroenterologist, but t1 > t0 for an oncologist. The authors state that physician specialty is not available to them, so I think that specialty needs more discussion in Limitations as an unmeasured confounder. 

9. My best understanding of the results is that lines 220-225 describe the analysis presented in table 4. However, lines 226-229 appear to describe a separate analysis, and it is not clear to me where these results are presented? 

Minor recommendations:

1. Abstract, line 49 (and other locations): the comparison of different exposure groups is implicit within retrospective cohort studies. Therefore, the term "two arm" is not needed, and confusedly may be taken to imply that this was a randomized study. 

2. Line 80: What does "pharmaceutical authorities" refer to? Drug companies? Regulatory bodies such as the FDA and the EMA? If the latter, "regulatory agencies" might be a better term. 

3. Line 90: the phrase "the fact that NIPMS seem to be published extremely rarely" does not make sense. See major recommendation #1 above regarding need for style and clarity edits. 

4. Line 99: "Error, bookmark not defined"

5. Lines 213-14: Authors state that trios without "at least one control" were excluded. Does this mean that those with only one control were still included? These would be pairs, not trios. Hence, "trio" might not be the best term if some of them were in fact 1:1 rather than 1:2. 

6. Line 221: Explain further. How were coefficients for matching trios generated? Was this a fixed-effects model, with trio-level effects? 

7. Table 3. How do the authors interpret the apparent increase in prescriptions among controls during t1? It does not seem to me that prescribing should increase at all in this group. 

8. Line 288: Does this statistic (1.04 with 95% CI 1.03-1.05) refer back to the first row of table 4? If so, why is this statistic described as an Rpr in one place and an OR in the other? If not, what table does it refer to? 

9. Throughout, it is not necessary to have sentences like "table X shows that…" Instead, it is clearer just to make the statement and then cite the referenced table at the end, as in (Table 3). 

10. Line 335-337. This explanation does not seem adequate. If exposed physicians were already more interested in the study drug in t0, then they would have been matched to higher-prescribing controls in t0. Matching would seem to have accounted for this. 

11. In discussion, consider comparing to findings by HE Glass, 2003: https://europepmc.org/article/med/15035835

12. The conclusion that NIPMS should be stopped is overly strong without a much fuller discussion of what the potential benefits and harms of these types of studies are. 

Reviewer #3: [Major Points]

1. This authors' perspective is very unique. The article is worth publication with the condition that the following are described clearly.

(1) Page7, Line128: What is the "Overall prescription volume"? Does it mean that a number of prescriptions per the doctor, per the drug, or per a certain period (e.g. per a month)? And what the aim of using it for control matching.

(2) Page7, Line133-137: As for Definitions of the "indicators of marketing use", the author just refereed their previous report but it should be described in this paper because these are necessary to interpreter the result accurately. I was not able to access the report on pub-med as probably that is written in German.

(3) Page9, Line171: What is the purpose of setting two periods of each four months for gathering the physicians' identified data?

(4) Page10, Line189: What are included in the "revenue generated by these prescriptions"? Is it only remuneration for the INPMS or included sales profits etc.?

(5) Page8, Table1, 3rd Cullum of Inclusion criteria for NIPMS: Dose the "follow up of one year (t2)" mean one year immediately after the end of the INPMS period? If not, which period has acquired?

(6) Page8, Table1, 7th Cullum of Inclusion criteria for NIPMS: Author have set the "approved at least 6 months before the beginning of the NIPMS" but 12 months were required to be satisfied one year data before INPMS (t0).

(7) Page13, Table2: There are several NIPMS whose duration is over one year. What rules were applied to extract one-year prescription data from one-year or more data?

(8) S1 Appendix. Selection of alternative drugs: How has the author identified alternative drugs when there was more than one drug meets the ATC-code criteria? If specified alternative drugs for each NIPMS were stated in S2 Table Detailed information on characteristics of NIPMS, it would be helpful.

(9) S2 Appendix. Matching Methods: What the aim of taking into account the "similar defined daily dose (DDD)-prescription of the studied drug" for identifying controls? In my understanding it depends on drugs, not on physicians.

2. In the discussion, the bellow points should be considered.

(1) Table 3 shows that even the control has a higher number of packages and DDD on t1 than t0 and t2. What could explain this trend most reasonably?

(2) Table 4 shows that there is a statistical difference in t0, before INPMS. It seems to indicate that physicians participating in a NIPMS already had a preference for new drugs (studied drugs) over existing drugs (alternative drugs), regardless of experience on NIPMS.

(3) In S2 Table, there is a large variety of "Duration approval date until begin (mo)" on NIPMS. In general, a situation of using drugs is affected by the time after approval. How did it affect the results of this study?

(4) In S2 Table, No.1-6 NIPMS have less than 10 physicians. Is there any possibility that few physicians influence the results?

[Minor Point]

3. Page6, Line99: there is "Error! Bookmark not defined."

Reviewer #4: I commend the authors in examining the association of the effect of a potential marketing device disguised as a research study with drug prescription. The retrospective study was based on a fairly large number of physicians of nearly 7000 using what appears to a national claims database that found participation in the NIPMS was associated with higher rates of prescription of the studied medication in the periods during the participation in the NIPMS and the year after.

Some specific questions and issues to clarify:

1) In the introduction section- Participation in NIPMS. How do physicians actually "participate" in NIPMS? How are physicians typically "selected" to participate? This seem to imply the physicians participate in the design/analysis of the study as opposed to something more similar to be detailed (marketed) about the drug. What are the participants suppose to do and actually do? 

2) In the methods section- National Association of Statutory Health Insurance Funds. Readers may not be familiar with this. Is this the name of the national claims database or prescription database? Or the name of agency? I assume this database includes all physician prescription database or only containing a subset of physicians or prescriptions? It seems the database only provide the prescription information for physicians in private practice. What is the implication of this? Are most physicians in private practice or work in the hospital? Is this referring to prescriptions in out-patient versus in-patient settings?

3) In the methods section- Study design on matching to controls. It is unclear to me (or at least that I can easily decipher) after reading through the methods section what criteria that authors used to match the participants in NIPMS and control. Matched on overall prescription volume at T0? What are included in the prescription volume? Could the patient's specialty impact the prescription volume? Matched to DDD and packages? Can the number of years out of training/medical school be found? Table 3- what is the prescription rate normalized to?

4) In the Methods section- NIPMS/ Table 2. The drugs studied is a very heterogenous class of drugs. Some have alternative equivalent. Some not as much (i.e. ivabradine). Some are much more prevalent/commonly used. In the analysis consider whether categories of class of drugs (cardiac, oncologic, ENT/primary care) affect the association of participation of NIPMS and prescription rate.

5) Analysis section of outcome measures- Have authors considered measuring "differences in difference" in the prescribing rates of the participants and controls in between the periods T0, T1, and T2?

[LINK]

---

## [Decision Letter · Decision Letter 1]

6 May 2020

Dear Dr. Koch,

Thank you very much for re-submitting your manuscript "Effect of physicians’ participation in non-interventional post-marketing studies on their prescription habits: A retrospective two-armed cohort study" (PMEDICINE-D-19-04307R1) for review by PLOS Medicine.

I have discussed the paper with my colleagues and the academic editor and it was also seen again by the original reviewers. I am pleased to say that provided the remaining editorial and production issues are dealt with we are planning to accept the paper for publication in the journal.

[LINK]

We look forward to receiving the revised manuscript by May 13 2020 11:59PM. 

Sincerely,

Clare Stone, PhD

Managing Editor 

PLOS Medicine

plosmedicine.org

Requests from Editors:

Title – please add a country setting and as below remove ‘effect’. I suggest a change from Effect of physicians’ participation in non-interventional post-marketing studies on their 2 prescription habits: A retrospective two-armed cohort study 

To

Impact of physicians’ participation in non-interventional, post-marketing studies on their prescription habits: A retrospective two-armed cohort study in Germany

Data – thank you for clarifying that the data is not publicly available. Please however add the URL /email or address or contact for those – who like you did – would like to approach the ministry of Health for access to the data. 

Abstract – please remove “Trial Registration: The trial was not registered, as it was not a clinical trial.”

As this is not a trial, please avoid saying "effect" (e.g. in the title; line 47; line 92; line 360 etc) based on this research design. "Impact" is the maximum that would be acceptable, I would think

Throughout manuscript - more careful about claims throughout, for example, you say "changes prescription rates" at lines 66 and 358 (however, all that is shown is that one group of physicians behave differently from a similar group)

Please briefly tell us about the classes of drugs studied in the abstract, e.g., X of 24 studies involved oncology agents, Y cardiovascular etc.

Is it possible to say something about the geography of this study – which cities? If too many to mention, maybe just a brief outline of some information – avoiding a list. Just to give a sense of relevance. 

You mention "patient harm" at line 97; I don't think there is any data on patient harm, is there? If so, I don't think you should imply that harm might result, rather saying, at most, "we have no data on patient benefits or harms".

STROBE checklist – please use sections and paragraphs instead of pages as these change during revisions and formatting, etc. 

Comments from Reviewers:

Reviewer #1: Thanks for the authors detailed response to my review. Their additional explanations are very helpful in helping clarifying some of my understanding, in particular how the authors minimised the effect of confounding for voluntary NIPMS participation by physicians.

In the response to this they have provided good justification on assessing matching criteria on the relative difference between cases and control during time period t0 as well as controlling for differences in the Poisson regression models. I think their approach is pragmatic and it would be strengthen their arguments, if they could detail this in their manuscript 

For the SS calculation - this is fine as a justification that they erred on a conservative approach to try to sample larger numbers as they felt a trial results may lead to under-powering their study. 

One final comment is that the results clearly show that there is increase in prescribing behaviour for the therapeutic products being involved in NIPMS, therefore it would most likely make sense that there might decrease rates in therapeutic products for the same indication which patients could be switched off from. Would the authors comment on whether they believe this effect could be observed?

Reviewer #2: Major comments:

1. Lines 186-188. Were the matching criteria (1) number of overall prescriptions in packages and (2) DDD of the studied drug and (3) DDD of alternative drugs? Or, (1) number of overall prescriptions in packages and (2) DDD of (the studied drug + alternative drugs)? Or something else? This should be clearer. Additionally, were both of these criteria (or all three, if that is the case) factored in equally to the matching? 

2. Related to #1 above, but also to my prior comment on the initial submission (Reviewer 2, minor recommendation #10), I am still not clear how, in a strictly mathematical/statistical sense, the t0 differences in prescribing presented in table 4 can be present if groups have been matching on t0 prescribing. If matching occurred based on (overall prescriptions) and (studied drug + alternative drugs), then it seems plausible that such a difference could arise if NIPMS physicians had at baseline a greater % of studied drug within the studied drug + alternative drug total. However, if matching used (overall prescriptions) and (studied drug) and (alternative drugs) separately, then it is not clear to me how the differences in table 4 persisted. Especially since it does not appear that the models were adjusted for other potential confounders, besides these prescribing measures? 

Minor comments: 

1. Line 120. I believe the authors are making a valid point about publication bias here. However, the phrasing of "NIMPS are also published extremely rarely" might be confused by the reader to mean "NIMPS are also conducted extremely rarely." I would suggest rephrasing to make it clear the NIMPS are often conducted but rarely published. 

2. Line 177-178. As phrased, it sounds as though the authors had to get approval from the individual NIPMS in order to get data on physician prescribing practices, which I do not expect was the intended meaning. 

Reviewer #3: The authors dedicated to clarifying the method of the study by revising it but there are still unclear points especially in the results. To help readers understand the manuscript without misinterpretation, I recommend additional revisions.

[Major Points]

1. Page 16, Line 307, "7-8% more of the studied drug during the NIPMS and 6-7% more during the year after the NIPMS" The figures have one digit after the decimal point, and how to calculated these percentages from the number of prescriptions in Table 3.

2. Page 18, Line 329, I'm not sure why the author examines only during t1 (NIPMS) not t2 (immediate after NIPMs) time frame for proportions of prescription of the studied drugs compared to alternative drugs. But this fact should be explicitly stated on page 20, line 362 of the discussion section.

3. Page 18, Table 5, The percentage in parentheses do not match the figures calculated by dividing each number of NIPMS by 24. Besides I don't understand what the author intended to explain from "not appropriate: 4 (14.8%); unclear whether appropriate: 6 (22.2%)."

4. Page 18, Table 5, The statements of marketing indicators should be described in the same terms as on the S4 Appendix and S2 Table.

[Minor Points]

5. Page 17, Table 3, I understood the periods are different among the t0, t1. If the average period of each time frame is shown in Table 3, it would be easily recognized the difference.

6. Before publication the author should look though details of the manuscript since there are several typos (i.e. lack of space; page 2, line 58, "NIPMSprescribed," page 16, line 304, "prescriptionsof").

[LINK]

---

## [Editor Report · Decision Letter 2]

27 May 2020

Dear Dr. Koch, 

On behalf of my colleagues and the academic editor, Dr. Sanjay Basu, I am delighted to inform you that your manuscript entitled "Impact of physicians’ participation in non-interventional post-marketing studies on their prescription habits: A retrospective two-armed cohort study in Germany" (PMEDICINE-D-19-04307R2) has been accepted for publication in PLOS Medicine. 

PRODUCTION PROCESS

PRESS

PROFILE INFORMATION

Thank you again for submitting the manuscript to PLOS Medicine. We look forward to publishing it. 

Best wishes, 

Clare Stone, PhD

Managing Editor 

PLOS Medicine

plosmedicine.org